# OpenReview forum: "Towards Cost-Effective Reward Guided Text Generation"
_ICML.cc/2025/Conference — ICML 2025 poster_

### Official Review · Reviewer_WyAk · 2025-02-21

**Overall Recommendation:** 4

**Summary:**

In earlier works on reward guided text generation (RGTG), the reward model is usually operationalized with a regression head on top of LM. However, this comes at the cost of having to evaluate all next possible tokens V times. This paper proposes a simple change that turns this into a V-channel head, just like normal language modeling head, to save cost.

**Claims And Evidence:**

Yes, they are supported by theoretical and experimental results.

**Essential References Not Discussed:**

NA

**Experimental Designs Or Analyses:**

Yes. I read the experimental results and they made sense. Most experimental set-ups follow from previous works to ensure comparability.

**Methods And Evaluation Criteria:**

Yes.

**Other Comments Or Suggestions:**

Yeah, after reading other reviewer's comments, I think a key point that's missing here is the scaling relationship between the size of the RM and the proposed method's performance.

**Other Strengths And Weaknesses:**

NA

**Questions For Authors:**

NA

**Relation To Broader Scientific Literature:**

NA

**Theoretical Claims:**

Yes. I went through the proofs.

---

> ### Author Rebuttal · Authors · 2025-04-01
>
> We want to thank you for reviewing our paper and your strong endorsement of our work.

---

### Official Review · Reviewer_cy1U · 2025-03-13

**Overall Recommendation:** 3

**Summary:**

This paper proposes an improved reward model for reward-guided text generation (RGTG), an alternative to offline RLHF for aligning language models with human preferences. Traditional RGTG incurs high inference costs as reward models score tokens individually and are optimized for full sequences, leading to suboptimal choices. To address this, the authors introduce a Bradley-Terry loss-based model that predicts optimal token expansions in a single step, reducing inference overhead. Theoretical analysis and empirical results show that the proposed method improves efficiency while maintaining competitive performance compared to existing RGTG and RLHF approaches.

**Claims And Evidence:**

S2: The FaRMA addresses the limitations of the existing RGTG method, which requires multiple calls of the reward model and suffers from unreasonable scoring for partial-sequences. The method significantly enhances efficiency while ensuring performance effectiveness;

S3: Theorems 1–3 provide a clear analysis of suboptimality in prior methods (PARGS, CD) and prove FaRMA’s guarantees under infinite training. This strengthens the paper’s credibility;

W3: While Theorems 1–3 highlight theoretical advantages, the paper does not analyze scenarios where FaRMA’s greedy token-wise optimization might diverge from global sequence optimality (e.g., compounding errors in long generations);

W5: In Section 4.2, the authors propose a novel approach that utilizes the maximization of V_θ (y_(1:i+1) |x) to determine V_θ (y_(1:i) |x) , thereby ensuring that V_θ (y_(1:i) |x) can represent the overall sentence-level reward. However, does this approach significantly amplify the computational cost of training the reward model, given that it requires iterating over all tokens in the sentence for the proposed constraint-based training? Have the authors considered experimenting with〖 V〗_θ (y_(1:i+k) |x) for ( k >= 1 ) ? Would such an approach offer a more efficient alternative while preserving the effectiveness of the method?

**Essential References Not Discussed:**

N.A.

**Experimental Designs Or Analyses:**

W1: The primary experiments in this paper utilize base models with 1B or 2.8B parameters. Additional Fine-Grained Text Generation experiments in the appendix, conducted with a 7B model, indicate that the performance advantage of FaRMA decreases as the base model size increases. Further empirical validation on this trend would enhance the evaluation of FaRMA’s robustness.

W2: The experiments primarily train reward models of the same size as the base model. If the reward model is always required to match the base model in size, FaRMA's advantage over DPO (which directly optimizes the base model) becomes marginal. Notably, in the HH Dialogue experiment, using a smaller reward model resulted in significantly worse performance. Exploring the influence of reward model size would clarify the limitations of the proposed approach;

W4: Narrow Baseline Comparison: CARDS is the only baseline compared under varying reward thresholds (Appendix D). A similar ablation for FaRMA’s hyperparameters (e.g., β in Eq.(14)) is missing, making robustness claims less substantiated.

**Methods And Evaluation Criteria:**

S4: The authors tackles the word ambiguity and complex network  problem for CNER task, which is benefit for the researchers that may encounter similar problems. Actually the word ambiguity problem of external knowledge is not easy to handle, this work points out a direction for it.

**Other Comments Or Suggestions:**

W6: In the introduction, the authors should ensure consistency in the use of punctuation between parentheses containing abbreviations and citation references.

W7: In Line:76, a period is missing before "On the TLDR."

**Other Strengths And Weaknesses:**

Strengths:
S1: Overall, this paper is well-written and easy to read.

S5: Results across diverse tasks (summarization, dialogue, UltraFeedback) show FaRMA outperforms RGTG baselines in reward scores while matching or exceeding RLHF methods (DPO/PPO) without LLM fine-tuning. The analysis of diversity (ROUGE-L) and GPT-4 preference rankings adds depth.

**Questions For Authors:**

Please refer to the above comments.

**Relation To Broader Scientific Literature:**

The authors tackles the word ambiguity and complex network  problem for CNER task, which is benefit for the researchers that may encounter similar problems. Actually the word ambiguity problem of external knowledge is not easy to handle, this work points out a direction for it.

**Theoretical Claims:**

W3: While Theorems 1–3 highlight theoretical advantages, the paper does not analyze scenarios where FaRMA’s greedy token-wise optimization might diverge from global sequence optimality (e.g., compounding errors in long generations);

---

> ### Author Rebuttal · Authors · 2025-04-01
>
> Thank you for taking the time to review our paper and numbering your comments.
>
> # W1:Performance Advantage on larger models
>
> We would like to point out the result on the 7 billion model (Table 5 Appendix) is on a different dataset, Ultra Feedback, and the absolute reward values on different datasets cannot be compared to each other. Additionally we observe that FaRMA on average performs better than the baselines including DPO. Therefore, based on the result we can not conclude that the advantage of FaRMa is diminished on larger models.
>
> Additionally we evaluate this setting using GPT-4 and provide the win-tie rate of FaRMA against DPO and ARGS for this experiment below:
>
>
>
> | Method A | Method B | Win-Tie % |
> | -------- | -------- | -------- |
> | FaRMA     | DPO     | 52      |
> | FaRMA     | ARGS     | 58      |
>
> We can observe that FaRMA has a higher winning rate and does well in this setting.
>
> # W2: Smaller Reward Model
>
> Please note that even at the same reward model size FaRMA has a computational advantage over DPO and PPO. Please refer to point 1 of the rebuttal for Reviewer wvoK. We note that FaRMA at the same model size trains $3\times$ faster. Therefore a smaller reward model would trade-off the performance against even more training efficiency.
>
> We also note that the smaller reward model is significantly better than $\pi_{ref}$ and comparable to RGTG baselines (at half the reward model size).
>
> # W3: Tokenwise Optimization and Optimality
>
> The best way to understand FaRMA is from a reinforcement learning (RL) perspective.  In RL, we train a policy to select actions that greedily maximize cumulative future rewards captured by the state-action value function.  Similarly, FaRMA chooses the next token greedily to maximize the value function.  The fact that FaRMA does not do a look-ahead search during decoding is consistent with RL since RL training ensures that the resulting policy should not need a look-ahead search during execution.  The look-ahead search is implicitly done during training (Eq. 17) by the estimation of the value function that captures cumulative rewards for all future steps.  Furthermore, the loss function in Eq. 17 naturally accounts for compounding of errors since the value of a partial sequence is trained with respect to the value of its continuations, including any error.
>
> # W4: $\beta$ Ablation for FaRMA
>
> | $\beta$ | r $\pm$ SE  |
> | -------- | -------- |
> | 0.5     | 1.33 $\pm$ 0.18     |
> | 1.0     | 1.77 $\pm$ 0.17     |
> | 1.5     | 2.11 $\pm$ 0.16     |
> | 2.0     | 2.1 $\pm$ 0.14     |
>
> We present an ablation on changing the value of $\beta$ for FaRMA on the TLDR dataset. A $\beta$ value of 1.5 is optimal in this case.
> We want to point out that the inference time is independent of $\beta$ for FaRMA. However, for CARDS the hyper-parameter controls the trade-off between higher reward and higher inference time. Therefore, we presented the result in the appendix to justify the threshold that we chose for CARDS.
>
> # W5: Computational Cost
>
> Our approach does not increase the training time since, in practice, instead of training over the entire sequence we sample some steps from the sequence. Note that this is standard in many ML algorithms, e.g. diffusion models [1].
>
> The sampling is done to keep the training set comparable to the preference dataset for full-sequence reward learning. Note that on Table 9 of the Appendix, on the same GPUs, the training time for ARGS (full sequence reward model) and Ours is comparable.
>
> Furthermore, your suggestion of matching the reward of a partial sequence to a longer trajectory instead of just the next token is an interesting variation. However, an exponential search in terms of vocabulary size is required to find such a trajectory with maximum reward, and this would lead to significantly higher training cost.
>
> 1. Bishop, Christopher M., and Hugh Bishop. Deep learning: Foundations and concepts. Springer Nature, 2023. Algorithm 20.1
>
> # W6 and W7: Typos and Corrections
>
> Thanks for pointing these out. We will correct these in our paper.

---

### Official Review · Reviewer_wvoK · 2025-03-14

**Overall Recommendation:** 4

**Summary:**

The paper proposes a new method to do reward-guided text generation (RGTG) that prefers the optimal expansion of a sequence and needs only one call to score all candidate tokens simultaneously. The experiments show that the proposed reward model leads to faster inference than other RGTG methods and performs on par with previous RGTG and offline RLHF methods.

**Claims And Evidence:**

The claim in 2nd Paragraph of Introduction, “Reward models are cheaper to train compared to offline RLHF updates even if both the reward model and LLM have the same number of parameters”. Even given the footnote associated with the statement, I’m still confused about why reward models must be cheaper than DPO? If the used reward models have the same number of parameters and the same output size (as this paper’s proposed method). In such case, one also need to load and make calls to one additional model (the reward model), which has the same size as $\pi_{ref}$. Do you actually assume that the training reward model requires fewer iterations than training the language model here?

I would suggest to revise the statement of “In RLHF, $\pi_{ref}$ is the quantity that we seek to improve so it does not make sense to improve $\pi_{ref}$ with a value function that depends on $\pi_{ref}$ itself” to something like “In RGTG, …..so it can introduce bias to train a value function depends on $\pi_{ref}$”. Since in more general RLHF, $\pi_{ref}$ may not mean the generation part, and using term like “introduce bias” could be more specific of prior work’s cons compared to “does not make sense”.

**Essential References Not Discussed:**

Beyond section 5 paragraph 3, there is much work discussing the fine-grained rewards or Q value functions when using RL to train language models. For example, “Adversarial Learning for Neural Dialogue Generation, 2017” proposed a method to learn the RL rewards (cast as a discriminator) by sampling the prefixes and updating the model with policy gradient. “Improving Conditional Sequence Generative Adversarial Networks by Stepwise Evaluation, 2018” and “Proximal Policy Optimization and its Dynamic Version for Sequence Generation, 2018” together approximate the Q-value for each prefix when learning the full episode RL returns (casted as discriminator) and update the model with PPO.

**Experimental Designs Or Analyses:**

For the experimental setup, how are responses paired to GPT-4 to score? Within one pair, are the two responses generated from different models? Or is one of them always the human-written response? If both responses are model generation, why can Figure 2 compare them all together? How many pairs are evaluated?

The 2nd paragraph in Section 6.2 mentioned again that “DPO and PPO based RLHF that is expensive to fine-tune”, however, there is no comparison of the total training time of the proposed FaRMA method (of the reward model) and the total training time of DPO (of the language model). The training equations of FaRMA itself when using a same size of the language model seem to possibly require a similar fine-tuning time. If not, it would be great to show the empirical evidence.

Line 416-419 on Page 8 mentioned that “we can further reduce the cost for both training and inference by reducing the reward model size while still improving over $\pi_{ref}$”. However, the experiment only demonstrates the same size and half size results on HH Dialogue, and the performance drops a lot from 1.80 to 1.41 average reward. If making this statement, I would expect to see the multiple sizes results (e.g., also include 1/8 and 1/8).

For the diversity experiment in Table 4, I’m interested in (1) the PARGS and CD results, as they are taken as the key baselines and discussed in Section 3, and (2) the temperature of sampling here, as temperature can impact a lot to the diversity.

**Methods And Evaluation Criteria:**

The method is reasonable for improving inference efficiency and the reward model performance, especially if the goal is to reward a prefix that can potentially grow into the optimal sequence, but not the average possible sequences a prefix can potentially achieve. However, for LLM, this method may need more proof that most inputs can find an optimal sequence and that there is no need to consider the output diversity.

**Other Comments Or Suggestions:**

* In page 7 line 354, “which is” is a typo that should be revised to clarify the whole sentence. I guess the authors mean all the baselines and the proposed method are sampled using top-k.
* In page 8 line 417, “can further reduces” => “can further reduce”.

**Other Strengths And Weaknesses:**

No, I have listed the strengths and weaknesses above.

**Questions For Authors:**

Among the above comments, I will change my evaluation if
1) the training time comparison of DPO and FaRMA is provided and FaRMA can be trained much faster than DPO as the paper claims,
2) the GPT-4 evaluation details (asked above) are provided and the setup is reasonable,
3) the diversity experiments expand with PARGS and CD, and
4) more prior work of models that score partial sequences are discussed (as the whole paper, including Abstract and Introduction continue talk about “the reward model is usually only trained to score full sequences” but much prior such contributions are not discussed.)

**Relation To Broader Scientific Literature:**

A new, more efficient method for reward guided text generation (RGTG) for LLM.

**Theoretical Claims:**

The three theory results, proved by counter examples, seem correct to me.

---

> ### Author Rebuttal · Authors · 2025-04-01
>
> Thank you for your detailed review and questions. We hope that our response will satisfy your concerns.
>
> # 1. FaRMA Efficiency vs DPO
>
> DPO is more expensive to train because it loads and makes calls to an additional reference model, $\pi_{ref}$, along with the model which is trained, $\pi_\theta$. FaRMA does not require access to $\pi_{ref}$. This also increases the memory footprint of DPO.
>
> | Method | Training time (mins) |
> | -------- | -------- |
> | FaRMA     | 82     |
> | DPO     | 254     |
> | PPO     | 238     |
>
> We compared the training time of PPO, DPO and FaRMA, on the TLDR dataset, on the llama3.2-1B model, on a single A100 GPU (to ensure fair comparison). As shown in the table above, FaRMA trains three times faster compared to DPO.
>
> | Method | Peak Memory (GB) |
> | -------- | -------- |
> | FaRMA     | 8     |
> | DPO     | 28     |
> | PPO     | 30     |
>
> We also looked at the peak memory usage per training batch for each of these methods. Again FaRMA has a much lower memory footprint.
>
> # 2. GPT-4 Evaluation
>
> The evaluation shows the win-rate of baselines vs FaRMA, i.e. DPO vs FaRMA, CARDS vs FaRMA, etc. This comparison is standard in literature [1,2,3]. We average this result over a hundred prompts and use the same hundred prompts for all comparisons. These prompts are randomly sampled, initially, and then fixed for all evaluations.  All the comparisons on the graph, including win-rate and inference time, are with respect to FaRMa. The blue point for FaRMA is at the $50\%$ mark on the y-axis for reference. We will add a horizontal line on the plot for an easier comparison.
>
> 1. Khanov, Maxim, Jirayu Burapacheep, and Yixuan Li. "ARGS: Alignment as Reward-Guided Search." ICLR 2024
> 2. Li, Bolian, et al. "Cascade Reward Sampling for Efficient Decoding-Time Alignment." ICML 2024 Next Generation of AI Safety Workshop.
> 3. Rashid, Ahmad, et al. "A Critical Look At Tokenwise Reward-Guided Text Generation." ICML 2024 Workshop on Foundation Models in the Wild.
>
> # 3. Diversity result for CD and PARGS
>
> **HH Dialogue Dataset**
> | Method | Rouge-L  |
> | -------- | -------- |
> | FaRMA     | 0.21 $\pm$ 0.02     |
> | PARGS     | 0.22 $\pm$ 0.02     |
> | CD     | 0.24 $\pm$ 0.01     |
>
> **TLDR Dataset**
> | Method | Rouge-L  |
> | -------- | -------- |
> | FaRMA     | 0.24 $\pm$ 0.01     |
> | PARGS     | 0.33 $\pm$ 0.01     |
> | CD     | 0.32 $\pm$ 0.01     |
>
> We present additional diversity experiments on CD and PARGS and observe that FaRMA still produces the most diverse text. The temperature used for all diversity tests is 1.
>
> # 4. Additional References
>
> We thank the reviewer for pointing us to these works. We will add them to our paper. The three suggested papers [1,2,3] train Generative Adversarial networks for dialogue generation. They employ either the policy gradient method [1,2] or PPO [3] to train the generator, and train the discriminator to provide rewards. To mitigate the problem of sparse rewards they employ methods of training step-wise Q-functions.
>
> However, we would like to clarify that the discussion in the abstract and introduction of our paper is in the context of RLHF, specifically reward guided text generation, that aligns LLMs to preference data at inference. Whereas the aforementioned works explicitly apply RL techniques to train text generators, RGTG methods avoid the use of off-line RL and instead employ reward guided decoding.
>
> 1. Tuan, Yi-Lin, and Hung-Yi Lee. "Improving conditional sequence generative adversarial networks by stepwise evaluation." 2019 IEEE Transactions
> 2. Li, Jiwei, et al. "Adversarial Learning for Neural Dialogue Generation." EMNLP 2017
> 3. Tuan, Yi-Lin, et al. "Proximal policy optimization and its dynamic version for sequence generation." arXiv preprint(2018).
>
> # 5. Optimal Sequence
>
> In FaRMA the search for an optimal sequence is done during training of the value function V.  While we could do a search for an optimal sequence during decoding, this is not desirable since this would increase decoding time.  Note that the search for an optimal sequence is implicitly achieved by the loss function in Eq. 17.  This loss function ensures that by the end of training the value of a partial sequence corresponds to the max of the values of all continuations.   This loss function is similar to the temporal difference loss function in traditional RL to estimate the value of the best plan going forward.   Hence at decoding time, the LLM does not need to search for an optimal continuation and does not worry about average continuations since the resulting policy is trained to select tokens that are greedy with respect to the value function, which already accounts for the best continuation.
>
> # 6. Typos and Corrections
>
> Thank you for pointing these out. We will correct them in the final version.

---

> > ### Comment · Reviewer_wvoK · 2025-04-05
> >
> > Thank the authors for the detailed response. While most of my concerns are addressed, I would like to confirm the following 2 questions before adjusting my evaluation.
> >
> > 1. Regarding the FaRMA efficiency vs DPO, I am curious about the reward model training time for FaRMA. Can you provide the time? I suppose your listed 82min is only the time using FaRMA with a trained reward model? Please correct me if the 82min already includes the reward model training time.
> > 2. I haven't seen a response to the following question in my review
> > >
> > Line 416-419 on Page 8 mentioned that “we can further reduce the cost for both training and inference by reducing the reward model size while still improving over”. However, the experiment only demonstrates the same size and half size results on HH Dialogue, and the performance drops a lot from 1.80 to 1.41 average reward. If making this statement, I would expect to see the multiple sizes results (e.g., also include 1/8 and 1/8).
> > >
> > To make it clear and correct my typo, I meant that could you provide the results of other reward model sizes, such as 1/4 and 1/8, to make the statement?

---

> > > ### Author Response · Authors · 2025-04-07
> > >
> > > Thank you. We are glad that we addressed your other concerns. Regarding these questions:
> > >
> > >  ## Q1: Training time FaRMA
> > >
> > > Please note that this table lists the FaRMA reward model training time. We are comparing the DPO training time with the reward model training time. Both are initialized from the llama 3.2 1 billion model.
> > >
> > > ## Q2: FaRMA with smaller reward model
> > >
> > > We present FaRMA results on the HH dataset with smaller reward models. These experiments were done on the Pythia language model and these are the available smaller models from this series.
> > >
> > >
> > >
> > > | Column 1 | r $\pm$ SE |
> > > | -------- | -------- |
> > > | $\pi_{ref}$ - 2.8b| 1.18 $\pm$ 0.12|
> > > | FaRMA - 400m | 1.49 $\pm$ 0.12|
> > > | FaRMA - 1b | 1.56 $\pm$ 0.18|
> > > | FaRMA - 1.4b | 1.41 $\pm$ 0.16|
> > > | FaRMA - 2.8b | 1.80 $\pm$ 0.18|
> > >
> > >
> > > All the FaRMA results are presented with the same $\beta=1.2$ which was used in the paper. We claimed in the paper that smaller FaRMA models still improve over $\pi_{ref}$. We note that even at $\frac{1}{7}^{th}$ of the orginal size, the FaRMA results are better than $\pi_{ref}$. We also note that even though the 1.4 billion FaRMA result is lower than the 1b and the 400m, these results are still within the standard error.
> > >
> > > Please let us know if you have any other questions.

---

### Decision · Program_Chairs · 2025-05-01

**Decision:**

Accept (poster)

**Comment:**

This paper proposes a method to reduce the complexity of reward-guided text generation, by estimating rewards for all potential tokens simultaneously, with better partial-sequence reward evaluation. Estimating per-token rewards across the whole vocabulary is not particularly novel, but the approach to make these work well at partial-sequence intervals is promising. Reviewers had a generally positive view of the paper. The main suggestions were to clarify some of the language of the claims and add quantitative support (especially for comparing efficiency / computational cost, relative to the choice of RM), to add evaluation details and more baselines, and to compare with more prior work. Some of these were addressed in the discussions, and these should find their way to the paper. Additionally, Theorem 3 (the only analysis for the proposed approach) uses the expression "optimal sequence" in terms of reward and not in terms of the actual decoding (Algorithm 2) of the method. Therefore, the claim that it supports empirical performance of the latter should be more conservatively tempered.